# A Systematic Review of PET Textural Analysis and Radiomics in Cancer

**DOI:** 10.3390/diagnostics11020380

**Published:** 2021-02-23

**Authors:** Manuel Piñeiro-Fiel, Alexis Moscoso, Virginia Pubul, Álvaro Ruibal, Jesús Silva-Rodríguez, Pablo Aguiar

**Affiliations:** 1Molecular Imaging and Medical Physics Group, Radiology Department, Faculty of Medicine, Universidade de Santiago de Compostela, 15706 Santiago de Compostela, Spain; manuelpifiel@gmail.com (M.P.-F.); alexis.moscoso@rai.usc.es (A.M.); alvaro.ruibal.morell@sergas.es (Á.R.); pablo.aguiar.fernandez@gmail.com (P.A.); 2Molecular Imaging Research Group, Nuclear Medicine Department, University Hospital and Health Research Institute of Santiago de Compostela (IDIS), 15706 Santiago de Compostela, Spain; virginia.pubul.nunez@sergas.es; 3Fundación Tejerina, José Abascal, 40, 28003 Madrid, Spain

**Keywords:** PET, radiomics, heterogeneity, textural analysis, cancer

## Abstract

Background: Although many works have supported the utility of PET radiomics, several authors have raised concerns over the robustness and replicability of the results. This study aimed to perform a systematic review on the topic of PET radiomics and the used methodologies. Methods: PubMed was searched up to 15 October 2020. Original research articles based on human data specifying at least one tumor type and PET image were included, excluding those that apply only first-order statistics and those including fewer than 20 patients. Each publication, cancer type, objective and several methodological parameters (number of patients and features, validation approach, among other things) were extracted. Results: A total of 290 studies were included. Lung (28%) and head and neck (24%) were the most studied cancers. The most common objective was prognosis/treatment response (46%), followed by diagnosis/staging (21%), tumor characterization (18%) and technical evaluations (15%). The average number of patients included was 114 (median = 71; range 20–1419), and the average number of high-order features calculated per study was 31 (median = 26, range 1–286). Conclusions: PET radiomics is a promising field, but the number of patients in most publications is insufficient, and very few papers perform in-depth validations. The role of standardization initiatives will be crucial in the upcoming years.

## 1. Introduction

Decades of research on cancer biology have revealed that tumors are heterogeneous entities at all scales (macroscopic, physiological, microscopic and genetic) [1,2,3,4], with different regions showing distinct morphological and phenotypic profiles [5,6,7]. Nowadays, it is widely accepted that tumor heterogeneity has profound implications in tumor development, therapeutic outcomes and survival [8,9,10,11], making it essential to develop methods for studying tumor heterogeneity in vivo [12]. 

In this context, non-invasive imaging techniques, such as magnetic resonance (MR), computed tomography (CT) and positron emission tomography (PET) become relevant due to their ability to provide information on the whole tumor in one acquisition [13]. Nowadays, imaging is central to cancer management, having applications in screening, diagnosis, staging, prognosis and treatment response, among other things [14,15,16,17,18]. Mainly, PET has emerged as the predominant imaging modality, overperforming conventional imaging techniques for the evaluation of blood [19], head and neck [20] or lung cancer [21]; and it has been increasingly proposed as an ideal tool for characterizing the biology of tumors at the macroscopic scale [13,22,23,24,25]. Over the last years, there has been an increasing interest in extracting quantitative information from PET images using image analysis [26,27]. Thus, semi-quantitative parameters such as the standard uptake value (SUV), the metabolic tumor volume (MTV) or the total lesion glycolysis (TLG) [28] obtained from ^18^F-fluorodeoxyglucose PET (FDG-PET) images, have been demonstrated to provide relatively objective information useful for the diagnosis, earlier evaluation and monitoring of treatment response [24,25,28]. These parameters are now fully incorporated into clinical guidelines and commonly measured at most hospitals in developed countries [29]. 

Several research studies have pointed out that high-order textural features derived from PET images can provide information about tumor heterogeneity, expanding the information available from clinical reports, laboratory tests and genomic or proteomic assays [26,27,30]. This has led to the incorporation of PET imaging to radiomics, a new medical imaging field exploiting image features to develop novel diagnostic, predictive and prognostic multiparametric models to support personalized clinical decisions and improve individualized treatment selection [27,31,32] (Figure 1**)**. Textural analysis has long been applied in CT [33] and MRI [34], but it had not been introduced in PET until the last decade. Since then, increasing numbers of studies have suggested that PET textural features would be correlated with tumor biology and heterogeneity [35,36,37,38,39,40,41,42], providing valuable information for tailoring individual treatments [13,18,23,43,44,45,46].

Despite the promising early reports, numerous studies have highlighted the challenges to be addressed before the use of PET radiomics becomes reliable and interpretable [46]. In addition to issues common to PET imaging itself, such as noise or partial volume effects [47,48,49], radiomics must deal with standardization issues related to the differences in acquisition and reconstruction parameters, post-processing techniques, tumor segmentation methods or even texture calculations [46,50,51,52,53]. The complex formulation of radiomics makes it challenging to explain pervasive findings such as the correlations displayed between different texture indices [54,55,56,57,58,59,60], as well as strong correlations between textural indices and tumor volume [61,62,63], which compromise the added value of these parameters in comparison with SUV or MTV. Furthermore, the variability in definitions and nomenclature of heterogeneity metrics themselves complicate any evaluation and comparison of published results [64]. Finally, there is no consensus about how many patients are needed, which textures must be extracted, or what methodologies should be applied for proper validation [24,46,52].

Due to the prominent role of quantitative image analysis in the diagnosis, tumor characterization and prognosis of cancer patients, the development of reliable and well-validated image analysis methods is of paramount importance. Particularly, parallel with the popularization of PET radiomics, several authors have highlighted the methodological and statistical issues in their works. In the present work, we provide a systematic review on the topic of PET radiomics, with a special emphasis on the applied methodologies and validation strategies.

## 2. Experimental Section

We computed this systematic review from 1 September to 15 October 2020. This review’s reporting complies with the PRISMA-P Preferred Reporting Items for Systematic Reviews and Meta-Analyses statement [65].

### 2.1. Search Strategy/Eligibility Criteria 

We performed a comprehensive literature search to identify eligible articles in the PubMed database searching for the presence of the combination of terms “(PET) AND ((radiomics) OR (texture) OR (textural))”. Results were admitted from 1 January 1990, up to and including 15 October 2020. The included articles satisfied all the eligibility criteria given in the subsequent paragraphs.

The inclusion criteria were (1) articles that included (in all fields) at least two of the search words specified in the search string; (2) studies based on human data specifying at least one tumor type; and (3) PET image was included.

The exclusion criteria were as follows: (1) studies not within the field of interest, i.e., not related to medical imaging, or applying PET textural analysis to diseases other than cancer, (2) preclinical and or animal studies, (3) studies including only testing data (i.e., phantoms, simulated data), (4) studies including less than 20 patients (for studies including several types of cancer, the articles were included if they provided data from 20 or more patients of at least one type), (5) case reports, reviews, poster presentations, conference abstracts and expert opinion papers, (6) articles including only first-order PET features in their analysis (SUV, MTV, TLG, or histogram features) and (7) articles not written in English. 

### 2.2. Data Collection, Selection Process and Items

After applying the inclusion and exclusion criteria, a review-specific electronic database (Microsoft Excel) was generated to handle document collection, data extraction forms and disseminate findings during all the study phases (abstract screening, eligibility and evaluation). As a final step, the full text of the included articles was evaluated.

From each article, the following information was extracted and annotated in the database:Cancer of body organs or systems, such as blood, brain, breast, gynecological, head and neck, liver or lung. Articles including patients of several cancer types were included in each category separately.Number of patients in a study (for studies evaluating several types of cancers separately, the number of patients for each was included and evaluated under the corresponding category).Radiotracer in use.Imaging modalities included in the radiomics analysis: PET, PET+CT, PET+MRI, PET+CT+MRI, other (note that not using PET for radiomics was an exclusion criterion).Type and number of imaging features extracted: first-order features (intensity: SUV and histogram features, shape or volume) and high-order features (textures). For the high-order features, the radiomics matrices used for feature extraction were annotated (grey-level co-occurrence matrix (GLCM), grey-level size zone matrix (GLZSM), grey-level run length matrix (GLRLM) or other). We also annotated if works used wavelet processing for data augmentation.Objective defining whether the article is focused on diagnosis/staging, prognosis/treatment response or tumor characterization. Publications aiming at the evaluation of technical factors were included in a separated category.Level of statistical validation evaluated with an ad-hoc scale. We ranked it poor in case of the absence of statistical analysis, average if statistics and radiomics were available on a single cohort (i.e., ROC analysis, Cox regression), good when findings were corroborated on a separate subsample and very good if validation against an independent cohort was available (i.e., from a different center/scanner).We annotated if the analyzed work found first-order or high-order features useful for the defined objective when available. This information is not present in every paper, as some of them only report the number of features in the final model without detailing the features included after processing. Furthermore, when methodologies such as deep learning are applied, this information is not be available.

### 2.3. Data Analysis

Full-text reviews were performed by two of the authors independently (M.P.F. and J.S.R.). Afterwards, discrepancies in their interpretation were identified and solved by consensus between all the authors. An initial assessment of the number of articles on each cancer group was made to select the most representative tumor types. We provide a separate section for those representing more than 5% of the gathered papers in the Results. The rest of the articles were included in a single group labelled as “Other”. 

### 2.4. Quality Assessment 

The same two reviewers performed quality assessment of the included studies. The assessment was performed using the ROBVIS tool (https://www.riskofbias.info/, accessed on 10 February 2021). The studies were assessed for three items: The number of patients, where studies with fewer than 50 patients were considered to have a high risk of bias; between 50 and 100 patients, some concerns; and more than 100 patients, low risk of bias. This rationale was adapted from the recommendations provided by Gillies et al. [31].Risk of overfitting, where studies with fewer than three patients per evaluated features were considered to have a high risk of bias; between three and five, some concerns; and more than five patients per features, low risk of bias. This rationale was adapted from the recommendations provided by Papanikolaou et al. [66].Level of statistical validation, where studies providing poor validation according to the scale above were considered to have a high risk of bias; average validation, some concerns; and good” or very good validation, low risk of bias.

Moreover, the overall score was given using the mode of the three punctuations. If the three scores were different, the overall score was “Some concerns”.

## 3. Results

### 3.1. Search Selection

The search process (Figure 2) returned a total of 744 publications. Three duplicated records were identified and removed. Titles and abstracts of the remaining 741 records were screened. After screening, we excluded 386 records due to inclusion/exclusion criteria. Full texts of the remaining 355 records were inspected. Sixty-five articles were excluded due to inclusion/exclusion criteria or incomplete information after a detailed assessment. Finally, a total of 290 full-text articles were included in the review. A complete list of the evaluated publications along with the data extracted from each publication can be found in the Appendix A. Using this selection, we corroborated the increasing pace of publications in PET radiomics in the later years. According to our findings, the first papers on PET radiomics appeared in 2009–2012 (one publication per year), and the number of publications started to rise after 2013 (eight publications), with nine publications in 2014 (+13% interannual increase), 19 in 2015 (+111% interannual increase), 20 in 2016 (+5% interannual increase), 44 in 2017 (+120% interannual increase), 46 in 2018 (+5% interannual increase), 66 in 2019 (+43% interannual increase), and 74 publications in 2020 (up to October 15) (+12% interannual increase) (Figure 3).

### 3.2. Data Analysis

#### 3.2.1. Database Characterization

Figure 4 shows the distribution of the evaluated publications by cancer types. In the following sections, we provide detailed data for cancer types representing more than 5% of the publications, namely: lung (71, 24%), head and neck (44, 15%), breast (20, 7%), gynecological (19, 7%), blood (18, 6%) and brain (17, 6%) cancer. The rest of the publications were synthesized in a category labelled as “Other”. Articles focusing on technical factors were evaluated independently. A meta-analysis of the whole database can be summarized as follows:The average number of patients included was 114 (median = 71; range, 20–1419). A considerable number of studies (187, 64%) included fewer than 100 patients.The average number of high-order features calculated per study was 31 (median = 26, range, 1–286). Most papers combined high-order with first-order features such as SUV (97%), volume or shape features (91%) and intensity histogram features such as kurtosis or skewness (76%).The most common matrices employed for texture calculation were the GLCM (included in 95% of the studies), GLZSM (63%) and the GLRLM (58%).Most of the studies included features only from PET (76%), followed by those studies combining PET and CT features (18%) or PET with MRI (16, 5%).Regarding the PET radiotracer, almost all the studies were performed using FDG data (91%) or combining FDG with another tracer (2%). Only a small number of studies (6%) were focused exclusively on other tracers.The most common study objective was prognosis/treatment response (46%), followed by diagnosis/staging (21%) and tumor characterization (18%). A total of 15% of the studies were dedicated to the analysis of technical factors. Figure 5 shows the distribution of objectives for the six most common cancer types in the meta-analysis.Most of the studies presented average (60%) or good (32%) levels of validation, and only a small number of studies performed proper validation using independent cohorts (8%).

#### 3.2.2. Quality Assessment

Out of the 290 publications evaluated, 109 were judged to have a low risk of bias, 131 were judged as having some concerns, and the remaining 50 were at a high risk of bias. A synthesis of the results for the three evaluated items is shown in Figure 6. The complete evaluation can be found in the Appendix A.

### 3.3. Main Types of Cancer

#### 3.3.1. Lung Cancer

We collected 71 articles evaluating lung cancer [37,57,58,67,68,69,70,71,72,73,74,75,76,77,78,79,80,81,82,83,84,85,86,87,88,89,90,91,92,93,94,95,96,97,98,99,100,101,102,103,104,105,106,107,108,109,110,111,112,113,114,115,116,117,118,119,120,121,122,123,124,125,126,127,128,129], of which 57 (80%) were focused on non-small-cell lung cancer (NSCLC). The average number of patients per study was 157 (median = 102, range, 22–1419) and the average number of studied textural features was 33 (median = 27, range, 1–286). GLCM was the most employed feature calculation matrix (92%) followed by GLRLM and GLSZM (50–55%). Eleven studies (15%) used wavelet filtering in the preprocessing for augmenting input feature numbers. A total of 97% of the publications reported that the inclusion of high-order features improved the performance of previous models. Although most of the studies used PET data only (62%), a relatively large subset (38%) used a multimodal approach combining PET and CT features. The radiotracer was FDG for all the studies. Regarding validation, 49% of the studies were classified as providing average statistical evidence, while 35% were classified as good and 11% as very good. 

The objective was diagnosis/staging in 20% of the publications, tumor characterization—in 25% of the publications, and prognosis/treatment response—in 54% of the publications. Especially relevantly, Kirienko et al. [79] compared the ability of PET and CT radiomics for differentiating primary and metastatic lung lesions in a pool of 482 patients, finding that only PET features were predictive. Other studies that focused on diagnosis demonstrated the ability of PET radiomics to distinguish between malign and benign FDG-avid lesions [67,71,74] or between tuberculosis and lung cancer [75,80]. Regarding characterization, in a study including 867 patients, Han et al. [129] showed that PET radiomics in combination with deep learning was able to differentiate histological subtypes of cancer, particularly, adenocarcinoma and squamous cell carcinoma. Other studies showed the ability of PET radiomics to predict EGFR mutation status [68,95,111,115] or PD-L1 expression [127]. As for prognosis, Arshad et al. [125] developed a prognostic model for the risk stratification of NSCLC patients, which was validated in several independent cohorts, including different scanner models and reconstruction protocols. The authors used first-order metrics, several texture matrices and wavelet filtering to get up to 665 features per tumor and concluded that a feature set independent of known prognostic factors could predict survival after radiotherapy/chemoradiotherapy. In contrast, in a recent study that included two independent cohorts, Konert et al. [126] concluded that PET radiomic features did not have a complementary value in predicting overall survival compared to conventional metrics.

#### 3.3.2. Head and Neck Cancer

We collected information from 60 articles that satisfy our selection criteria and evaluated head and neck cancer [28,82,130,131,132,133,134,135,136,137,138,139,140,141,142,143,144,145,146,147,148,149,150,151,152,153,154,155,156,157,158,159,160,161,162,163,164,165,166,167,168,169,170,171,172,173,174,175,176,177,178,179,180,181,182,183,184,185,186,187]. The average number of patients was 126 (median = 72, range, 20–707), and the average number of calculated textural features was 33 (median = 24, range, 1–98). The GLCM was the most common matrix for feature calculation (93%), followed by the GLSZM (62%) and the GLRLM (55%). Most studies combined textural features with first-order statistics (93% of the studies included SUV, 85%—shape/volume, 82%—histogram features), and eight studies (13%) used wavelet filtering for multiplying feature numbers. Forty-four studies included only PET data, fourteen studies (23%) combined PET and CT data, and two studies (3%) combined PET and MR data. FDG was the most common radiotracer (93%), but some studies also included data from ^18^F-fluoromisonidazole (FMISO) (3%) and ^18^F-fluorothymidine (FLT) (3%). Most studies provided average validation 60%), followed by good (26%) and very good validation (13%). Regarding the objectives, 10% of the studies evaluated diagnosis/staging, 17% of the studies focused on tumor characterization and 73% of the studies evaluated prognosis/treatment response. 

Among the most relevant works, Du et al. [132] evaluated the ability of radiomics to differentiate between recurrence and inflammation in nasopharyngeal carcinoma, obtaining an AUC of 0.89. The method was validated in an independent subsample of the cohort. Choi et al. [150] evaluated the association between the tumor–stroma ratio and heterogeneity features. They found that coarseness (a feature from the neighborhood gray tone difference matrix (NGTDM) matrix) correctly evaluated this tumor characteristic (AUC = 0.741). Finally, regarding prognosis, Vallières et al. [182] used sophisticated machine learning strategies for developing a model predicting treatment outcome from FDG-PET and CT images of 300 patients. The model was developed using data from two different centers and validated independently on two additional cohorts. The developed model could predict locoregional recurrences and distant metastases with AUCs of 0.69 and 0.86, respectively.

Similarly, Peng et al. [187] developed a model for selecting those patients that would benefit from induction chemotherapy. Sörensen et al. [145] investigated whether textural features on FMISO PET before chemoradiotherapy could identify patients with better overall survival, concluding that higher homogeneity of tumor hypoxia could correlate with a better outcome and open the door to novel applications of radiomics on novel tracers. Nevertheless, this study was performed in a limited cohort of 29 patients, and further evaluations including more patients are needed. 

#### 3.3.3. Breast Cancer

We evaluated 20 publications on breast cancer [59,188,189,190,191,192,193,194,195,196,197,198,199,200,201,202,203,204,205,206], all of them using FDG as the radiotracer. The average number of patients was 118 (median = 81, range, 34–435). The average number of calculated textural features was 18 (median = 6, range, 2–73), extracted mainly from the GLCM (19/20, 95%), the GLSZM (9/20, 45%) and the GLRLM (7/20, 35%). All the evaluated papers included SUV, volume/shape, and histogram features. A total of 75% of the studies included PET data alone, while 20% combined PET and MR radiomics and one study (5%) combined PET and CT. Fifteen studies (75%) fell in the category of average validation. In comparison, five studies (25%) were considered to provide good validation and 20% of the evaluated works focused on diagnosis/staging, 35%—on tumor characterization and 45%—on prognosis. 

In one of the most relevant works assessing diagnosis, Ou et al. [194] evaluated the ability of PET and CT radiomics to differentiate breast carcinoma from breast lymphoma using a machine-learning approach. They validated their findings in a separate subsample of the cohort, obtaining an AUC of 0.81 for PET radiomics, which outperformed CT radiomics. Nevertheless, in a similar work, the same authors showed that this result might represent minimal improvement to the classification performance obtained by SUV alone [192]. In Moscoso et al. [203], the authors showed that PET textural features are correlated with immunohistochemical factors and the immunohistochemical subtype of breast cancer using images acquired using a dedicated breast PET scanner. Finally, Lee et al. [188] developed a statistical model combining clinicopathological factors and texture parameters from PET and CT images to predict individual responses to neoadjuvant chemotherapy.

#### 3.3.4. Gynecological Cancer

We collected 19 publications focusing on gynecological cancers [82,104,207,208,209,210,211,212,213,214,215,216,217,218,219,220,221,222,223], particularly on cervical (74%), endometrial (16%), epithelial (5%) and vulvar cancer (5%). The average number of patients was 93 (median = 84, range, 20–190). The average number of calculated textural features was 33 (median = 31, range, 2–73). The most used heterogeneity matrices were the GLCM (17/19, 89%), followed by the GLRLM (15/19, 79%) and the GLZSM (14/19, 74%). All the publications used FDG as the radiotracer. Regarding modalities, 16 studies (84%) used PET data only, and three studies (16%) combined PET with MRI. All the evaluated papers combined textural features with SUV metrics, while 18 studies (95%) included volume/shape measurements, and 14 studies (68%) included first-order metrics derived from the intensity histogram. A total of 79% of the studies found that radiomic features were useful for the pursued objective, in most cases (60%) accompanied by first-order metrics. Three studies (16%) reported that only first-order metrics were predictive. As for the level of statistical validation, nine studies fell into the average validation category (47%), eight—into the good category, and two (11%)—into the very good category. A total of 42% of the publications focused on diagnosis/staging, 11%—on tumor characterization, and 47%—on prognosis/treatment response. 

Li et al. [211] evaluated the ability of PET radiomics to predict pelvic lymphatic metastases in patients with early-stage cervical squamous cell carcinoma. While the authors reported that some GLCM features might have value in predicting the vascular endothelial growth factors (VEGF), they found that the best staging was obtained when using TLG combined with histogram metrics. In a similar work, Shen et al. [212] concluded that homogeneity from the GLCM combined with TLG was the best combination of predictors, partially supporting these results. In a study involving 84 patients, Novikov [221] evaluated whether PET radiomic features of epithelial tumors would be able to predict the differentiation grade, reporting an accuracy between 91% and 100%. Finally, regarding prognosis, Lucia et al. [223] presented one of a very few papers aiming at validating a previously developed radiomics model [222], providing compelling evidence of the ability of PET+MR radiomics to predict disease-free survival and locoregional control in locally advanced cervical cancer.

#### 3.3.5. Blood Cancer

We collected 18 papers about radiomics in blood cancers [82,224,225,226,227,228,229,230,231,232,233,234,235,236,237,238,239,240], most of them focusing on non-Hodgkin lymphoma (11/18) and Hodgkin lymphoma (4/18). The average number of patients included was 77 (median = 51, range, 24–251) and the average number of evaluated textures was 28 (median = 22, range, 3–78), extracted mainly from the GLCM (100%), the GLZSM (56%) and the GLRLM (50%). All the reviewed studies combined high-order features with some first-order parameters, such as volume or shape features (100%), SUV (94%) and histogram-derived parameters (50%). All the publications used FDG. Regarding modalities, 72% of the studies explored PET radiomics, while the remaining 28% combined PET and CT textures. For the level of validation, 12 publications qualified as average (67%), five—as good (28%) and one—as very good (3%). Of the studies, 55% reported that combining first-order parameters and high-order textures provided the best performance for the pursued objective, while 22% reported that the best models used only first-order parameters and 33% obtained the best results when using only high-order features. Regarding objectives, 34% of the papers focused on diagnosis/staging and 66%—on prognosis/treatment response. 

Among the most relevant publications, Mayerhoefer et al. [226] investigated FDG-PET radiomics as an alternative to biopsy to assess bone marrow involvement in mantle cell lymphoma. They found that the developed radiomic signature combining PET textures with laboratory data had an AUC of 0.81 predicting bone marrow involvement with reasonable accuracy. Milgrom et al. [233] assessed whether radiomic features extracted from baseline PET scans predicted relapsed or refractory disease status in a cohort of 251 patients with stage I–II Hodgkin lymphoma and found that a model incorporating SUV, MTV and three high-dimensional features was able to predict primary refractory disease with a model AUC of 0.95. The validation was carried out on a subset of the cohort not included in the training process. In contrast, Jamet et al. [231] developed a model for predicting transplant eligibility in newly diagnosed multiple myeloma using 139 patients from two independent cohorts. They concluded that a combination of SUV and clinical parameters is the best and most robust predictor after validation on an independent sample. 

#### 3.3.6. Brain Cancer

We collected 17 publications [225,241,242,243,244,245,246,247,248,249,250,251,252,253,254,255,256] focusing on the evaluation of brain tumors, most of them gliomas (71%). The average number of patients was 71 (median = 70, range, 20–127) and the average number of evaluated textures was 40 (median = 33, range, 2–75), extracted mainly from the GLCM (94%), the GLZSM (65%), and the GLRLM (65%). All the studies combined textural features with SUV, volume or shape and histogram metrics. The most widely reported modality was PET (used in 76% of the studies), while 18% of the studies combined PET with MR and one study (6%) included PET, MR and CT features. Aside from FDG, which was included in 39% of the studies, six publications (33%) included data using ^18^F-2-fluorotyrosine (FET), three (18%)—using ^18^F-methionine (MET), one (6%)—using ^18^F-fluorodopa (FDOPA) and one study—using FMISO radiotracers. Regarding the level of validation, 71% of the studies provided good validation, 24%—average” validation and 6%—very good validation. Regarding the objectives, 65% of the papers focused on diagnosis, targeting issues such as differentiating recurrence from radiation injury [244,245,246,248], while 35% of the studies evaluated tumor characterization. 

Kong et al. [255] studied the correlation between radiomic features and proliferative activity in primary gliomas as measured by Ki-67 in 123 patients. The authors conclude that the developed radiomic signature could stratify patients into two distinct prognostic groups, with results comparable to those obtained with Ki-67. Li et al. [256] developed an FDG-PET radiomic model for predicting the isocitrate dehydrogenase (IDH) genotype and prognosis and obtained AUCs > 0.9 for both the training and validation patient datasets. Finally, Qian et al. [253] developed a model for predicting the MGMT methylation status in glioblastoma patients using FDOPA images and obtained accuracy of around 80%. The validation was carried out in an independent subsample of their dataset. 

#### 3.3.7. Other Cancers

We collected 47 publications on “Other” cancer types. This category included studies on gastrointestinal (11) [257,258,259,260,261,262,263,264,265,266,267], pancreatic (8) [268,269,270,271,272,273,274,275], sarcoma (8) [276,277,278,279,280,281,282,283], neuroendocrine (5) [284,285,286,287,288], prostate (4) [289,290,291,292], thyroid (3) [293,294,295], thymic (2) [296,297], skin (2) [298,299], liver (2) [300,301], and renal carcinomas (1) [302]. The average number of patients was 84 (median = 70, range, 26–214) and the average number of textural features extracted was 29 (median = 17, range, 1–236). In a similar manner to previous cases, GLCM (44/47, 94%) was the most common feature matrix, followed by GLSZM (29/47, 62%) and the GLRLM (26/47, 55%). Regarding modality, 42/47 studies (89%) used only PET data, while 2/47 studies (4%) combined PET and CT and 3/47 studies (6%) combined PET and MR data. The most common radiotracer was FDG (38/47, 81%), but the presence of edotreotide (DOTATOC)/DOTA-octreotate (DOTATATE) radiotracers was relevant in the papers evaluating neuroendocrine tumors (4/5, 80%). Prostate-specific membrane antigen (PSMA) radiotracers (^68^Ga or ^18^F) were used for all papers focused on prostate cancer (4/4, 100%). A total of 19% of the studies were focused on diagnosis/staging, 23%—on tumor characterization, and 57%—on prognosis/treatment response. Regarding the validation process, 36/47 studies (77%) presented average validation, while ten fell into the good validation category. Forty-four out of forty-seven papers (94%) concluded that radiomic features provided predictive value for the pursued objectives, in most cases (33/44, 75%) in combination with first-order features. 

#### 3.3.8. Technical Factors

Finally, we found 44 papers evaluating technical factors [53,60,61,62,64,154,303,304,305,306,307,308,309,310,311,312,313,314,315,316,317,318,319,320,321,322,323,324,325,326,327,328,329,330,331,332,333,334,335,336,337,338,339], without pursuing a clinical objective. Of these, 30% evaluated the impact of different tumor segmentation methods. Among these, Hatt et al. [317] assessed the robustness of PET textural features among the segmentation methods with and without partial volume correction and provided a selection of features robust to these parameters. A total of 22% of the studies evaluated the impact of different acquisition and patient management protocols with a particular interest in the effects of respiratory motion. In this regard, Grootjans et al. [325] investigated the impact of respiratory motion and noise on textural features, concluding that respiratory motion significantly affected PET textures’ quantification. Only 14% of the papers evaluated the impact of different reconstruction methods and parameters, and 9% of the studies assessed the correlation between textural features and conventional parameters such as SUV and MTV. Hatt et al. [62] found a significant correlation between heterogeneity features and MTV and observed that the textures’ complementary information increased progressively with volume. Finally, 11% of the studies proposed and validated harmonization strategies and 14% of them evaluated other factors’ influence. 

## 4. Discussion

### 4.1. Summary of the Main Findings

The present work presents a comprehensive overview of the available literature in PET radiomics and textural analysis. We extracted detailed information from 290 articles after data curation according to the PRISMA-P methodology. We observed that the interest in PET radiomics increased exponentially in the last decade, and we expect that the current tendency will continue in the upcoming years.

Lung, head and neck, breast and gynecological cancers emerged as the most studied cancer types. It is well-known that radiomics is a demanding technique in terms of data, and thus we assume that this is a consequence of the prevalence of these types of cancer [340]. The average number of patients per study was 114, and we observed that 65% of the evaluated publications included data from less than 100 patients (29%, less than 50 patients). Previous works evaluating radiomic methodologies suggested a proportion of five patients per evaluated feature to avoid model overfitting [52,66] or a minimum of 100 patients [31]. Based on this and considering that the average number of high-order features was 31 (median = 26, range, 1–286), we would recommend including larger numbers of patients in further works. In addition to this, a limited number of publications (8%) provided a validation of the proposed models using independent databases or a validation in an independent subsample of the initial dataset (28%). We assume that this fact is related to the small amount of data available. Publications with higher numbers of patients used resources such as The Cancer Imaging Archive (TCIA) [341]. We expect that the continuous growth of public imaging databases will improve these numbers in the future and provide a common playground where algorithms could be compared and validated. The rise of collaborative models such as federated learning [342,343] where different centers share models validated across centers while keeping their own data anonymous will also generate more robust algorithms in the near future.

Furthermore, we observed that very few papers aimed at externally validating previously developed models [128,223]. We believe that this should be a common practice in the field. Nevertheless, we observed that most of the publications did not provide easy access to the developed models. As for data sharing, accessibility of software is a substantial concern, reducing the applicability and potential impact of published studies and models [344]. The collaboration between centers and researchers should be improved for further model validation, which is needed to generalize PET radiomics. In this regards, van Griethuysen et al. recently released PyRadiomics, an open modular platform to provide the community with standard tools for promoting further independent development and evaluation [345]. 

Regarding the objectives of the evaluated publications, the evaluation of prognosis or treatment response was the most common (46%), followed by diagnosis and staging (21%), tumor characterization (articles aimed at predicting the biological characteristics of tumors traditionally obtained by other means (i.e., histology, genetics)) (18%) and the evaluation of technical factors (15%). This result is not surprising, as textures are intended to measure heterogeneity, well-known to be related to aggressiveness and poor outcomes [177,178,179]. We believe that more papers focused on tumor characterization and technical factors would be useful in this regard. On the one hand, although heterogeneity itself was proposed as a biomarker in the past [346,347] it has been reported that heterogeneity is related to biological characteristics such as tumor microenvironment [348,349], genetic expression [350] and, macroscopically, tumor grades or cancer subtypes [351]; thus, linking radiomic features with particular biological characteristics could provide additional evidence for the field. On the other hand, further work is still needed to evaluate the impact of image acquisition, reconstruction, post-processing and feature calculation [352]. In an exemplary work, Bodowicz et al. investigated an association of PET radiomics with local tumor control after radiochemotherapy in head and neck cancer, developing radiomic implementations with two different software packages. Of the 649 features calculated, only 12% were reproducible between the two software implementations, and, although both models were similarly predictive, they included different sets of features, pointing to the need of further harmonization on feature calculation. Harmonization initiatives, such as the Image Biomarker Standardization Initiative [64], would be extremely helpful for this process, as reliable and reproducible measurements are of paramount importance for biomarkers to progress on their validation. 

Most of the evaluated studies (78%) concluded that the inclusion of textural features improved the results of models developed with clinical or first-order imaging metrics, which is a powerful conclusion. Nevertheless, it must be interpreted with caution since, as mentioned above, most of the evaluated studies included an insufficient number of patients or were not extensively validated. Many studies reported models including both first and high-order features, which were compared usually with counterparts including a reduced number of features, so an improvement of the result can be caused by overfitting [353,354]. Several papers from our group [63] and others [60,126,317] have suggested a strong correlation between the most used textural features and conventional parameters such as MTV and SUV, which must be untangled.

In summary, PET radiomics is an up-and-coming field, and PET radiomics might have a role in clinical practice in the future. In this regard, the results of the most relevant papers are very appealing. Nevertheless, we have identified several methodological concerns related to the validation of the purposed algorithms, the number of patients included, the lack of data and software accessibility and a need for further methodological standardization. We have also observed that the community is already developing solutions to overcome these limitations.

### 4.2. Limitations

Our review presents several limitations. We selected studies with at least 20 patients for statistical reasons, which could cause the risk that tumors with low prevalence (or novel radiotracers) were systematically excluded from our analysis. Furthermore, we decided to divide the analyzed articles into six main groups, with one other group collecting all the cancer types representing less than 5% of the publications. This categorization might hide different approaches for different cancer types in this last group or among subtypes in the other groups. In addition to this, several papers investigated the application of radiomics to sites different from the primary tumor, such as those applying textural analysis to lymph nodes. These different approaches are not considered in the present review. Several publications did not fit our objective’s classification as they evaluated more than one topic. In such cases, classifications were decided on consensus. We assume that this might systematically exclude secondary topics common to a high number of publications. Furthermore, we decided to report the total number of high-order features calculated by the authors instead of those found to be predictive/useful. This decision was made to better reflect the high heterogeneity in methodology across studies. Moreover, information about the number of useful features or how many of them were used was not available in many studies. Finally, although we obtained some information on the publication results, a more detailed analysis, potentially focusing independently on each type of cancer, is needed to assess, for example, if certain features are repeatedly found predictive for certain diseases.

## 5. Conclusions

The bibliography available on the topic of PET radiomics is exponentially increasing. Although most papers presented promising results, we found that the methodology was highly variable. Furthermore, we observed that the number of patients included in most publications was insufficient and that very few papers performed in-depth validations. Based on the obtained data, we can conclude that PET radiomics is a promising field in its early days of development, and we expect that the interest in PET radiomics will continue growing in the following years.

## Figures and Tables

**Figure 1 diagnostics-11-00380-f001:**
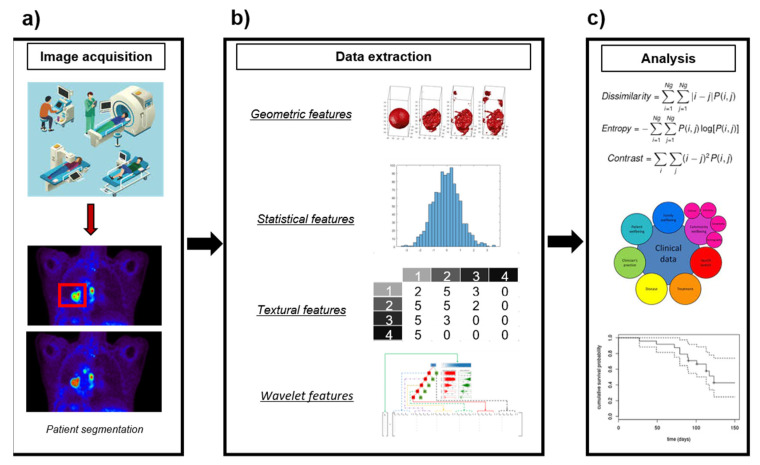
Example of image processing for radiomics. (**a**) Image acquisition and tumor segmentation. (**b**) Extraction of different shape, intensity, and textural features from the segmented tumor. (**c**) Development of prediction models using imaging features.

**Figure 2 diagnostics-11-00380-f002:**
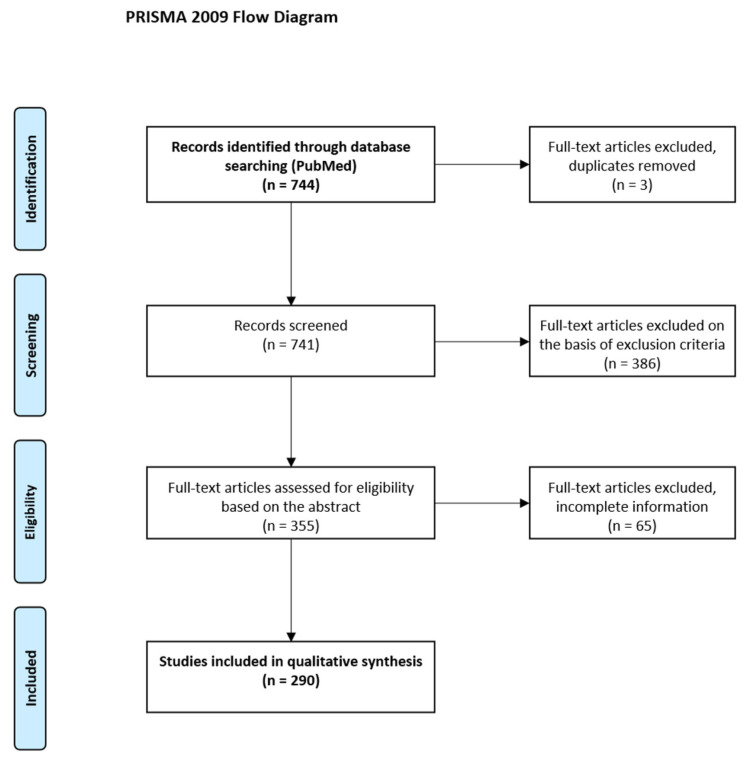
Preferred Reporting Items for Systematic Reviews and Meta-Analyses flow diagram.

**Figure 3 diagnostics-11-00380-f003:**
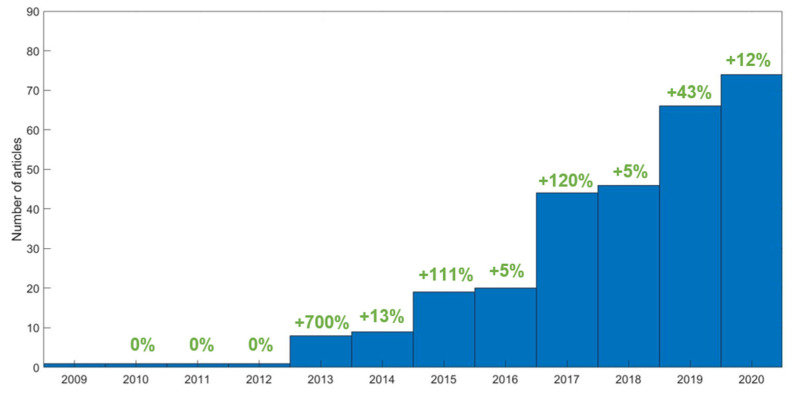
The number of publications per year. In green, the interannual increase.

**Figure 4 diagnostics-11-00380-f004:**
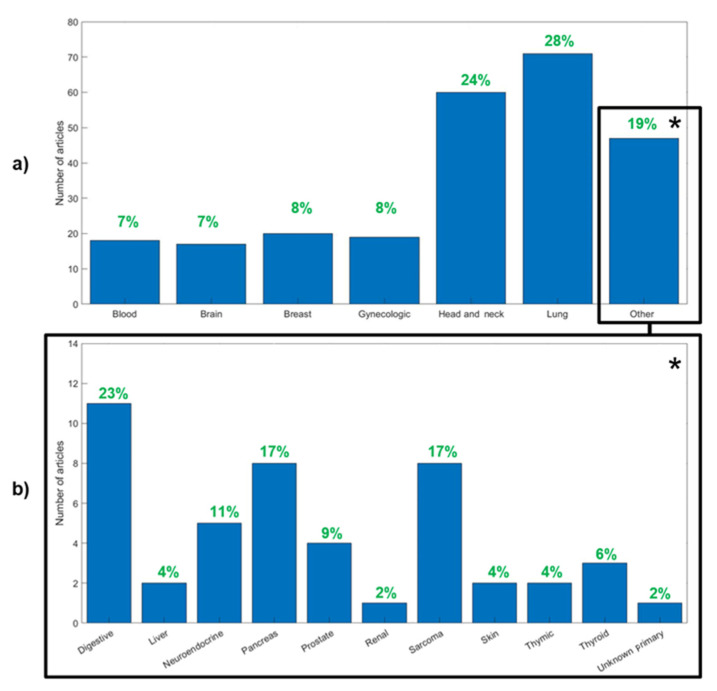
The number of articles associated with each cancer type group. (**a**) Number of patients per number of groups studied in the present review. (**b**) Distribution of cancer types in the “Other” group. In green, the percentage of publications corresponding to that group. * Percentages in panel (**b**) represent the percentage in the “Other” group.

**Figure 5 diagnostics-11-00380-f005:**
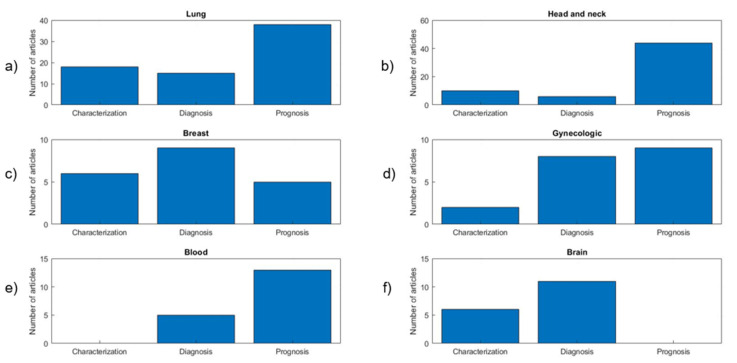
Studies targeting each of the defined objectives segmented by cancer type. (**a**) Lung, (**b**) head and neck, (**c**) breast, (**d**) gynecologic, (**e**) blood, (**f**) brain cancer.

**Figure 6 diagnostics-11-00380-f006:**
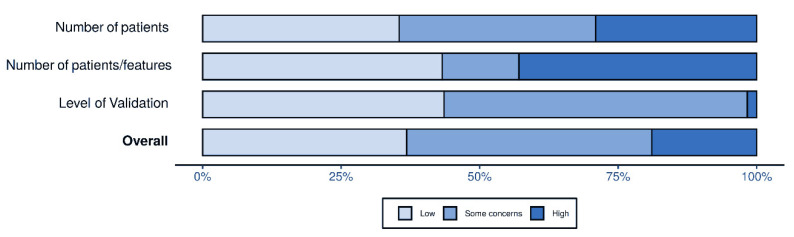
Per item and overall risk of bias.

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
