# Peer review of "A Systematic Review of PET Textural Analysis and Radiomics in Cancer"

_diagnostics, 2021, doi:10.3390/diagnostics11020380_

Round 1

Reviewer 1 Report

diagnostics-1092497

Title:A systematic review of PET textural analysis and radiomics in cancer.

General Comments:

This paper aimed to perform a systematic review of studies concerning the application of PET Texture Analysis and Radiomics in cancer patients. All selected studies are classified according to application fields (diagnosis, tumor characterization, and prognosis) and cancer object of the study. For each study were assessed the number of patients, features extracted, and level of statistical validation. 290 paper were selected, with the average of 114 patients included and 31 high-order features calculated in each study. Almost all papers combined high-order with first-order features, and the highest number of the studies are concerninglung tumor and prognosis.

Several studies showed some promising results, but their main limitations are in the methodology and application in a routine clinical setting.

The topic is interesting; however, the manuscript has several limitations, in almost all sections.

Please see specific comments

Specific comments:

Title: ok.

Keywords:Please modify the keywords, these seems too generic. You should highlight the application fields of TA, Radiomics and PET.

Abstract:

  • The “Background” section seems too generic, in particular you should introduce the challenges of PET Texture Analysis and Radiomics in oncology.
  • Please modify “Methods” section, you should briefly describe inclusion, exclusion criteria, and analysis performed (i.e. risk bias).
  • Please modify the “Results” section by adding the number of studies selected for each field of application. In addition, please rewrite the sentence “Most papers combined these high-order features with first- order features such as SUV (97%), volume or shape features (91%) and intensity histogram features such as kurtosis or skewness, etc. (76%).”, it seems overall confusing. Is it a systematic review or a metanalysis as describe in this section? Please clarify.
  • What about papers regarding cancer diagnosis, tumor characterization and prognosis? Please briefly add in “Results” section.
  • Please rewrite the “Conclusion” section, you should summarize your results and mention the lack of TA standardization.
  • Please reduce the use of “we expression”.

Introduction:

  • First paragraph is not proper related to the aim of the manuscript. Please shorten it and add some references.
  • Please rewrite the sentence “In addition to these simple, well-established parameters, research studies have pointed that high-dimensional parameters, known as textural features, can provide information about tumor heterogeneity, providing a phenotype complementary to that pro- vided by clinical reports, laboratory tests and genomic or proteomic assays”, it sounds quite unclear.
  • Please avoid in the text the term “as exemplified in Figure 1.” and you might replace with “(Figure 1)”. In addition, each panel should be labelled “a”, “b” and “c” and individually descripted in the caption. Please modify.
  • The sentence “While these approaches were rapidly adopted for CT [34] and MRI [35]” seems too reductive. In fact, Textural and Radiomics approaches lack of standardization and this limits their application in the clinical setting as routine.
  • Please rewrite the aim of the study, it seems too reductive. You should highlight role of quantitative analysis in diagnosis, tumor characterization, and prognosis of cancer patients.

Experimental Section:

  • Please be more consistent with the time frame in which you performed the research.
  • Please specify in a more specific manner the terms used for the research, it seems too reductive. Did you use the restriction language or not? Did you select only the more relevant paper or not?
  • Please extend inclusion and exclusion criteria, these seem too reductive. What about studies with overlapping populations? Please specify.
  • Please pay attention on exclusion criteria, you listed twice “reviews”.
  • Please rewrite the classification of articles reviewed, it sounds overall confusing.
  • Please avoid the expression “etc…” in the MM section.
  • Please rewrite the sentence “PET radiotracer/s, and the modalities in addition to PET from which features where extracted (PET, PET+CT, PET+MRI, etc.)”.
  • Please avoid the discussion of “Characterization” in this section, it should be moved in “Discussion” section.
  • Please define the expression “When possible” that you used in point 7 of “Data collection, selection process and items.” section.
  • Please expand the “Data analysis” section, it seems too reductive. What about the evaluation of methodological quality of included studies? How many reviewers performed the studies selection, data extraction and quality assessment? Please specify.
  • How did you evaluate the level of statistics validation? Please specify.
  • Please rewrite the “Risk of bias” section, it sounds unclear.

Results:

  • Please avoid the first paragraph, it is not necessary the description of the “Results” section.
  • Please modify the report of papers excluded, it seems overall confusing. In addition, please modify Figure 2, articles excluded should be listed in a clearer manner.
  • Please add the total increasing number of papers, for each year of analysis, also in the text.
  • You should consider modifying the description of metanalysis in a more concise manner.
  • Please structure each paragraph in the same manner: paper collection, validation, objective and discussion of the main relevant papers.
  • Please add some references in this section, in particular when you divide the paragraph for application field and describe the papers included.
  • Lung cancer: Please modify the order of studies’ objectives (P7, L248-249)by listing as first “diagnosis”, “tumor characterization”, then “prognosis”.
  • Head and neck cancer: You should mention the study in a proper manner (XX et al.) by avoiding the term “In [75]”. Please modify the sentence “Nevertheless, this study was performed in a limited cohort of 29 patients and findings should be corroborated in further studies.”, it seems to be not complete.
  • Breast: What about the main papers regarding diagnosis? In addition, please maintain the same order: diagnosis, tumor characterization and prognosis.
  • Gynecologic:Please add the description of the main relevant papers included for each objective of the papers, in concordance with the previous paragraph.
  • Blood:Please add the objective and the description of the main relevant papers included, in concordance with the previous paragraph. In addition, you should mention the reference [82] in a proper manner, please see comment above.
  • Brain: Please rewrite the paragraph in order to maintain the same structure for each paragraph. In addition, you should add the description of the main relevant paper included for each objective.
  • Others cancers: Please rewrite the paragraph in the same structure of previous paragraphs.
  • Technical factors: Please mention the references properly, see comment above. You should avoid the expression “for example” in Results section.

Discussion:

  • Please discuss your results, it seems too reductive the only listing without a proper discussion.
  • Please modify the sentence “it is clear that the number of patients included in most of the publications is scarce and insufficient for ensuring the validation of the developed radiomics signatures.”, it is a quite strong statement.
  • What about the technical problems in terms of texture parameters extraction, method of segmentation, and different PET specifications? Please discuss.
  • Please extend the discussion of the main fields of application, this section seems too reductive.

References: see specific comment above.

Figures: Overall, figures need some layout improvement.

  • 1 and 2: see above.
  • 3: Please modify the figure by adding the percentage of increasing for each year. In addition, you may rewrite the caption, it seems to be redundant.
  • 4: Please consider adding the percentage for each “other cancer” and modify the caption. You should label the two different figure panels with “a” and “b”, also the “*” used in the figure has to be described in the caption.
  • Figure 5: You should label the different figure panels with a different letter (i.e. “a” and “b”), please modify the figure and the caption.

Supplemental material: You should divide the papers selected according to different application fields.

Linguistic and typewriting: formatting andtypewriting revision are necessary.

Reviewer 2 Report

The authors present a review of the published article dealing with PET radiomics in cancer. The review is extensive and well organized. 

Author Response

The authors present a review of the published article dealing with PET radiomics in cancer. The review is extensive and well organized.

We thank the referee for the time expended reviewing our manuscript and the positive comments. Following the review process, we have procuded several improvements to our manuscript. We hope that the referee sees these modifications as positive.

Reviewer 3 Report

The paper is well written and the topic is  very interesting and relevant.

I would suggest to add at the beginning of each paragraph  the number of the references of the papers examined in the review- i.e. for lung tumours: ” We collected 71 articles evaluating lung cancer, of which 57 (80%) were focused on 237 Non-Small Cell Lung Cancer (NSCLC)” [56, 64-7].Concerning the description of findings obtained in blood cancer, the authors collected 18 papers, but they did not clearly indicate the type of hemato-oncological malignancy investigated.  This is particularly relevant as the number of the examined studies is small. The authors should add the number of the studies available  for each hemato-oncological disease, the type of malignancy (lymphoma? Melanoma?), the grade (high or low grade) and the site used for radiomics and textural analysis (lymph nodes? Brain for CNS lumphoma? Bulky mass?..).

In the Discussion the authors should add  their opinion on the role of textural analysis and radiomics in the clinical practice and the future aspects of the wide diffusion of the findings described in the study

Round 2

Reviewer 1 Report

diagnostics-1092497_R1

Title:A systematic review of PET textural analysis and radiomics in cancer.

General Comments:

Authors followed only in part the comments of the previous revision.

Some sections should be still modified and improved. In addition, a linguistic revision is mandatory.

Please see specific comments for further details.

Specific comments:

Keywords:

  • ok

Abstract:

Some of the changes were performed. Some adjustments are still needed in some aspects:

  • In Results section; please pay attention on percentage of objective of the study, the sum of these was not 100%.

Introduction:

  •  

Experimental Section:

Please expand the “Data analysis” section, it lacks of evaluation of methodological quality and quality assessment for included studies.

  • Please rewrite the sentence “PET imaging is used as an inclusion criterion, the presence of CT and MRI data is not representative of the proportion of works using CT and MRI radiomics.”, it sounds unclear.

Results:

  • In “Database characterization” please define if for each cancer type you report the number of papers included (i.e. lung (71)) or the percentage.
  • It is mandatory to add some references in this section, in particular when you divide the paragraph for application field and describe the papers included for objective. Please modify. An alternative might be adding tables with referral at least to type of tumor and application fields with a dedicated column with all the articles cited just by the corresponding reference.
  • Overall, you should structure each section in the same manner, by adding the main paper regarding each objective of the study (i.e. diagnosis, tumor characterization, and prognosis) for each application field. Please modify (e.g. head and neck section is only focused on response to treatment and survival).

Discussion:

  • This section has been correctly improved.

References: see specific comment above.

Figures:

  •  

Supplemental material:ok

Linguistic and typewriting:furtherlinguistic revision is necessary.

Round 3

Reviewer 1 Report

All the necessary changes has been performed.